# Metabolism of *Malus halliana* Roots Provides Insights into Iron Deficiency Tolerance Mechanisms

**DOI:** 10.3390/plants13172500

**Published:** 2024-09-06

**Authors:** You-ting Chen, Xia-yi Zhang, De Zhang, Zhong-xing Zhang, Yan-xiu Wang

**Affiliations:** 1College of Horticulture, Gansu Agricultural University, Lanzhou 730070, China; cyt198211@163.com (Y.-t.C.); zhangxy2020@lzu.edu.cn (X.-y.Z.); m18394189590@163.com (D.Z.); zzx19872354@163.com (Z.-x.Z.); 2Affairs Center of Jingtai County Forestry and Grassland Bureau, Baiyin 730900, China

**Keywords:** *Malus halliana*, iron deficiency, physiological, metabolomic

## Abstract

Iron (Fe) deficiency is one of the most common micronutrient imbalances limiting plant growth globally, especially in arid and saline alkali regions due to the decreased availability of Fe in alkaline soils. *Malus halliana* grows well in arid regions and is tolerant of Fe deficiency. Here, a physiological and metabolomic approach was used to analyze the short-term molecular response of *M. halliana* roots to Fe deficiency. On the one hand, physiological data show that the root activity first increased and then decreased with the prolongation of the stress time, but the change trend of root pH was just the opposite. The total Fe content decreased gradually, while the effective Fe decreased at 12 h and increased at 3 d. The activity of iron reductase (FCR) increased with the prolongation of stress. On the other hand, a total of 61, 73, and 45 metabolites were identified by GC–MS in three pairs: R12h (Fe deficiency 12 h) vs. R0h (Fe deficiency 0 h), R3d (Fe deficiency 3 d) vs. R0h, and R3d vs. R12h, respectively. Sucrose, as a source of energy, produces monosaccharides such as glucose by hydrolysis, while glucose accumulates significantly at the first (R12h vs. R0h) and third time points (R3d vs. R0h). Carbohydrates (digalacturonate, L-xylitol, ribitol, D-xylulose, glucose, and glycerol) are degraded into pyruvate through glycolysis and pentose phosphate, which participate in the TCA. Glutathione metabolism and the TCA cycle coordinate with each other, actively respond to Fe deficiency stress, and synthesize secondary metabolites at the same time. This study thoroughly examines the metabolite response to plant iron deficiency, highlighting the crucial roles of sugar metabolism, tricarboxylic acid cycle regulation, and glutathione metabolism in the short-term iron deficiency response of apples. It also lays the groundwork for future research on analyzing iron deficiency tolerance.

## 1. Introduction

Iron (Fe) is an important mineral nutrient that is vital for a variety of cellular and other physiological functions ranging from metabolism to growth and development. However, Fe availability is very low in calcareous soils, which is caused by high pH and poor aeration [1]. With respect to Fe acquisition, apples are referred to as strategy-I plants. The reduction strategy includes root morphological, physiological, and biochemical changes that lead to an increased capacity for Fe uptake. Morphological changes include root tip swelling, the development of transfer cells, and an increase in the number of lateral roots [2]. In apples grown under Fe deficiency, the ferric chelate reductase (FCR) and H^+^-ATPase activity are up-regulated in the plasma membrane [3], which promoted the reduction of iron to ferrous, decreased the pH value of rhizosphere, and increased the synthesis of organic acids.

At the metabolic level, increases in the activity of the glycolytic pathway and the tricarboxylic acid (TCA) cycle have been found in different plant species grown under Fe deficiency [4]. Sucrose, glucose, and fructose, as metabolites, are substrates for the synthesis of complex carbohydrates such as starch and cellulose. In addition, sugar provides the basis for the biosynthesis of amino acids and fatty acids and all other compounds in plants [5]. Glutathione (GSH) is probably one of the most important metabolites involved in the defense responses against environmental stresses [6]. It plays a pivotal role in the interaction between plants and symbiotic nitrogen-fixing bacteria, in the compartmentalization and neutralization of xenobiotics and heavy metals, and in the vacuolar transport of secondary metabolites [7]. GSH has also been suggested as the main donor of the reduced sulfur group for glucosinolate biosynthesis in Arabidopsis [8]. It is reported that some plants can accumulate and release both reducing and chelating substances, such as phenolics and flavins. Here, the Fe-deficiency-induced synthesis of phenolics is an important part of a plant’s adaptive strategy that encourages a reutilization of considerable amounts of Fe [9]. Certainly, the efficiency of a plant’s response to Fe deprivation in its natural environment should be determined by its capacity to activate several metabolic pathways, which as a whole contribute to solve the problem [10].

Calcareous soils account for approximately 30% of the world’s cultivated soils, and Fe deficiency is a widespread agricultural problem affecting crop yields [11]. Therefore, the selection of deficiency-tolerant rootstocks is a more beneficial way to aid the development of the apple industry. Among the discovered apple varieties, *Malus halliana*, as a native variety, is an excellent apple rootstock with strong drought resistance and salt tolerance, which is an important plant resource for breeding [12]. It is also one of the most important commercial rootstocks and scion materials in China [13]. As research progressed, Wang et al. [14] found that *Malus halliana* has a high tolerance to iron deficiency. Existing studies have investigated its salt alkali tolerance [15] through metabolomics and transcriptomics, as well as the transcriptomics of iron deficiency resistance [16] and some functional genes [17,18]. However, there is still no research on the metabolomics of iron deficiency in *Malus halliana*, especially in its roots. One of the reasons is that, under iron deficient conditions, plant roots find it difficult to absorb enough iron from the soil to maintain their normal needs. Therefore, studying the impact of iron deficiency on root systems is of great significance. Large scale metabonomics analysis is a tool to obtain a global view of plant response to abiotic stress. In this study, GC–MS was used to analyze the response of *M. halliana* roots to Fe deficiency stress and to explore the mechanism of Fe deficiency tolerance.

## 2. Material and Methods

### 2.1. Plant Materials

The experiment was conducted at the College of Horticulture, Gansu Agricultural University. Firstly, we disinfected the seeds of *M. halliana* from Gansu province, China, with 0.2% KMnO_4_ for 30 min, and then washed them with tap water for 12 h. Subsequently, the seeds were stored at low temperature in sand for 30–40 days, and, after completion, they ripened and germinated. Finally, they were sown in small pots. When the seedlings had 10 mature leaves, we transplanted them into a foam box containing half-strength Hoagland’s solution [19]. After one week of growth, we switched this to complete Hoagland’s solution. During this period, we controlled the indoor environment (16 h light/8 h dark, 22 ± 1 °C, 60–70% relative humidity) and replaced the nutrient solution every 5 days. When the plant grew new leaves (two weeks), we selected well-growing and uniform seedlings for treatment: CK (40 μmol·L^−1^) and -Fe (4 μmol·L^−1^). The roots were harvested after 0 h, 12 h, and 72 h of Fe-deficiency stress for metabonomics analysis, with 0 h serving as a control. Each biological replicate comprised ten plants. The time points were selected according to previous physiological experiments [10].

### 2.2. Determination of Root Physiological Parameters

FCR activity was determined according to Grusak [20] and Lei et al. [21], and some methods were modified in our study. The roots of *Malus halliana* and hydroponics seedlings were immersed in saturated CaSO_4_ solution for 5 min, and then the roots were rinsed clean by deionization. The roots of each treatment were placed in 50 mL 0.5 mM Fe (Ⅲ)—EDTA and 0.4 mM 2,2′-bipyridine nutrient solution without Fe. After reacting in the dark for 2 h, and the wavelength of the reaction solution was 520 nm. Then, the treated roots were weighed, and finally the formula of enzyme activity was brought in: FCR (nmol·g^−1^·h^−1^) = V × A_520_ × 10^9^/8650/M/T, where V is the volume of reaction solution, M is the weight of roots (g); T is the reaction time (h), and each experiment was repeated three times.

Dynamic monitoring of rhizosphere pH [22]: At 11:30 every day after the 7 d of Fe deficiency stress treatment, the pH value of culture medium was monitored by PHS–2C (Changzhou Pusen Electronic Instrument Factory, Changzhou, China) precision pH meter, the initial pH of nutrient solution was adjusted to 6.00, the electrode was close to the root surface during the measurement, and the nutrient solution was not changed during the monitoring period. Comparison of treatment and control of each variety was undertaken.

Root activity was analyzed by the triphenyl tetrazolium chloride (TTC) method [23]. Effective Fe is extracted by dilute hydrochloric acid, and all Fe is digested by dry ashing [24]. We used a WinRHIZO Pro LA2400 scanning electron microscope (Regent Ltd., Vancouver, BC, Canada) for observation and photography.

Seedlings subjected to control and Fe-deficient treatments at 0 h, 12 h, 3 d, 6 d, and 9 d were selected for measurement. The leaves of each seedling were removed and scanned using a scanner (EPSON Expression 11000XL, EPSON, Suwa City, Japan). The height of each seedling was measured using ImageJ image analysis software (v1.8.0). After measuring the height, the roots and stems of the plants were washed with deionized water, and the surface moisture was absorbed with filter paper. They were then placed in envelopes and killed at 110 °C for 15 min before being dried at 70 °C to a constant weight (DW). The underground and above-ground weights were measured, and the biomass of each individual plant was calculated. The calculation formula is as follows:Root to shoot ratio = root dry weight/aerial dry weight

Determine the content of nitrogen (N), phosphorus (P), and potassium (K) in roots using the method of Baoshidan [25].

### 2.3. GC/MS Analysis

The derived samples were analyzed on the Agilent 7890B gas chromatography system (Agilent Technologies Inc., Santa Clara, CA, USA) using the same method as Zhang’s experiment [10]. The mass spectrometer is injected at intervals (each “sample”) throughout the analysis run, with the aim of providing a set of data that can be evaluated for repeatability.

### 2.4. Data Preprocessing and Statistical Analysis

The data format was converted using ChemStation (version E.02.02.1431, Agilent Technologies Inc., Santa Clara, CA, USA) software. Metabolites are annotated by Fiehn or NIST databases. We conducted a Log2 transformation of the data in Excel 2007 (Microsoft, Redmond, WA, USA) (using 0.000001 to replace 0 before transforming) and then imported the data matrix into the SIMCA package (14.0, Umetrics, Umea, Sweden). Principal component analysis (PCA) and (orthogonal) partial least squares discriminant analysis ((O)PLS-DA) were used to analyze the differential metabolites.

### 2.5. Selection of Differential Metabolites

The selection of differential metabolites was based on the VIP values (>1.0) derived from the (O)PLS-DA model and the *p*-value (<0.05) of the two-tailed Student’s *t* test for the standardized peak area of different groups.

### 2.6. Statistical Analysis

Experimental data are presented as the mean ± sd of three independent replicates. Data were analyzed via one-way ANOVA (*p* < 0.05). SPSS (version 22.0, IBM, Armonk, NY, USA) was used for statistical analysis, and Origin 9.0 software was used for data processing (OriginLab, Hampton, MA, USA).

## 3. Results

### 3.1. Selecting the Time Point for Metabonomics Analysis in Roots of M. halliana Seedlings

To study the root growth of *M. halliana* under Fe deficiency stress, physiological parameters were measured at five time points. In this study, the root system of *M. halliana* was significantly changed under the Fe deficiency condition compared with the control (0 d). As shown in Figure 1a, with the prolongation of the stress time, the root activity first increased and then decreased, reaching the peak value on day three. However, the change trend of root pH (Figure 1d) was just the opposite, and it dropped to the lowest value on day three. The total Fe content decreased gradually, while the effective Fe decreased at 12 h and increased on day three (Figure 1b,c). The activity of iron reductase (FCR) (Figure 1e) increased with the prolongation of stress and remained unchanged after 3 d. Throughout the entire stress period, the plants exhibited significant changes both above and below ground. As the dry weight of the root system increased, the root to aerial also gradually increased, but the height difference was not significant (Appendix A). Therefore, Fe deficiency affects plant growth.

To determine the sampling time point, we observed changes in root morphology and element content. In Table 1, the total root length gradually increases with the increase in stress days. The trend of changes in root surface area and total root volume is the same as the trend of changes in the total number of roots. However, there was no significant difference in average root diameter at the five time points. Compared with the control group, the number of root tips and branches increased by 9.74% and 5.03%, respectively, after 9 days of stress. From the above series of root indicators, it can be seen that the plants have begun to undergo changes after 12 h of iron deficiency. In Appendix A, both potassium and phosphorus underwent significant changes in the short term, with opposite trends. In order to further investigate the short-term response of plants to iron deficiency changes, we conducted metabolic analysis at 0 h, 12 h, and 3 d to further explore their intrinsic changes.

### 3.2. Metabolic Trajectory of M. halliana Roots under Fe Deficiency Stress

The metabolic changes in the roots of *M. halliana* seedlings subjected to Fe deficiency treatments were determined through GC–MS. As shown in Figure 2, a multivariate statistical analysis was performed on the normalized data. The results of the identified metabolite (Figure 2 M1) show that, under Fe deficiency, there is a good separation in PCA and the data are feasible. The control and Fe-deficiency treatment samples in leaves were separated by the first principal component (PC1), representing 53.0%, 58.4%, and 56.3% (Figure 2 M2, M5, M8) of the total variation. There were significant differences between the two groups in the PLS-DA score chart (Figure 2 M3, M6, M9), and the model interpretation rate R2Y and prediction rate Q2 were higher (R2Y = 0.995, 0.978, 0.997; Q2 = 0.966, 0.827, 0.789). OPLS–DA analysis was used to maximize the differentiation of metabolic patterns at three time points. R2 values shown in the figure are 0.956, 0.986, and 0.914 (Figure 2 M4, M7 and M10), respectively, indicating that this prediction is stable and effective.

### 3.3. Comparative Analysis of the Metabolites of M. halliana Root System under Fe Deficiency

Under the stress of Fe deficiency, the metabolites of *M. halliana* changed significantly. In the Venn diagram, 61, 73, and 45 metabolites were found in three pairs R12h-R0h, R3d-R0h, and R3d-R12h, respectively. Meanwhile, a total of 42, 32, and 21 metabolites were identified between R12h-R0h vs. R3d-R0h, R3d-R0h vs. R3d-R12h, and R12h-R0h vs. R3d-R12h (Figure 3). Evidently, the comparison of these three data sets shows that there are 15 metabolites overlapping among the three time points, accounting for 15.2% of the total metabolites.

The OPLS-DA model was used to select differential metabolites according to a variable influence on projection (VIP) values and *p*-values (VIP > 1 and *p*-value < 0.05). Then, a hierarchical cluster was analyzed on the metabolites in each group of samples. A total of 451 peaks remained after filtering and denoising, and 176 metabolites were identified under Fe deficiency in *M. halliana* roots. The results showed that 33, 42, and 29 metabolites were up-regulated (FC > 1), and 20, 21, and 12 metabolites were down-regulated (FC < 1) among the differentially expressed metabolites (DEMs) during three time points, respectively (Figure 4, Appendix A). In R12h-R0d, the content of 2′-Hydroxyacetophenone (2,922,332.717 times) and m-cresol (5.088 times) were higher, and the Digalacturonic acid (2,167,125.539 times), lactic acid (5.421 times), succinic acid (2.354 times), and benzoic acid (2.305 times) were highly expressed in R3d-R0d and R3d-R12h, respectively. The expression levels of 3-(4-hydroxyphenyl) propionic acid (0.0948 times), Benzoylformic acid (0.228 times), N-Acetyltryptophan acid (0.116 times), Norleucine acid (0.157 times) dl-p-Hydroxyphenyllactic acid (0.0522 times), and Alizarin acid (0.147 times) were lower, respectively, in three time points of *M. halliana* roots.

### 3.4. Pathway Mapping and Metabolite-to-Metabolite Network Visualization

The KEGG enrichment analysis of different metabolites showed that 54, 56, and 55 metabolic pathways were accumulated (Appendix A), and bubble charts were made for the top ten pathways, respectively (Figure 5). The results showed that based on the value of −log*P* and the score of path impact, 10 pathways were, respectively, enriched with DEMs in three time points. In R12h-R0d (Figure 5a, Appendix A), the biosynthesis of alkaloids derived from ornithine, lysine, and nicotinic acid, as well as glyoxylate and dicarboxylate metabolism, and the citrate cycle (TCA cycle) enriched important metabolites. At the second time point, there were significant differences in the metabolic pathway, which included Alanine, aspartate, and glutamate metabolism; Pantothenate and CoA biosynthesis; and valine, leucine, and isoleucine biosynthesis in roots. Importantly, Alanine, aspartate, and glutamate metabolism in R3d-R0d and R3d-R12h was significantly enriched. The results showed that *M. halliana* responded positively to Fe deficiency through these pathways.

To further explore the response mechanisms of *M. halliana* against Fe stress, the differently expressed metabolites that participated were summarized on a simplified metabolic map in the various metabolic pathways (Figure 6). The metabolism of digalacturonate, L-xylitol, ribitol, D-xylulose, glucose, and glycerol (Figure 6) changed significantly at three time points. Those carbohydrates were degraded into pyruvate through glycolysis and pentose phosphate, which participate in the TCA; they are transformed into other amino acids, such as threonine, L-asparagine, and L-valine, etc., through glutamic acid. Interestingly, cytosine is metabolized by glutathione to produce other compounds. Like other metabolites, putrescine enters the TCA through butanoate metabolism to provide energy for plants. Meanwhile, the accumulation of nicotinurate and putrescine was significant under Fe deficiency, while the contents of oxalic acid and xylitol were down-regulated at R0d-R12h and up-regulated significantly at the last two time points. The results showed that the tolerance of plants to Fe deficiency was caused by the change of metabolites.

## 4. Discussion

### 4.1. Root System Changes of M. halliana Seedlings in Response to Fe Deficiency

Root morphological parameters are one of the main factors reflecting plant growth and development. In the case of Fe deficiency, the Fe-deficient genotypes in mechanism I plants showed the morphological changes of root growth speed, root tip swelling, thickening, and the formation of a large number of root hairs, and a large number of transfer cells were also formed in the outer skin cells and root hair areas of the roots of this explant, that is, the obvious characteristics of the morphological response of iron deficiency adaptability [26]. In this paper, we found that, with the extension of Fe deficiency stress time, the total root length and number of root tips and branches all increased. However, the average root diameter did not change significantly. Plants have evolved specific root responses, increasing the root system’s ability to absorb Fe and the proportion of available Fe in the rhizosphere [27], resulting in an increase in root dry weight and an increase in root–shoot ratio (Appendix A). It has been reported that, under Fe deficiency stress, the root of maize appears to have rod-shaped swelling, and the lateral root increases significantly [28]. Fan et al. [29] thought that the varieties with higher root number may have stronger tolerance to Fe deficiency. These morphological changes can increase the surface area of nutrient uptake by roots, so it is generally considered to promote the absorption of Fe by roots. Physiological changes such as root tip morphology and reduction capacity, proton outflow, and Fe absorption occurred simultaneously under Fe stress in sunflower [30]. In Figure 1c, with the extension of Fe deficiency time, significant changes in Fe content occur in *M. halliana,* and the total Fe content showed a downward trend. The most effective explanation is to supplement Fe by activating the absorption and transportation of Fe by roots to maintain the normal growth of plants. Differently, the active Fe decreased at 12 h and increased at 3 d (Figure 1b). It is estimated that at least 75% of the Fe in the root system is stored in bienfait under hydroponic condition, and, in the condition of Fe deficiency, this part of Fe can be effectively reused by plants [31]. Zha et al. [32] found that *M. xiaojinensis* has a high content of active Fe in its roots, which is due to the expression of genes related to Fe absorption and transport in its roots, which can also explain the increase in active Fe content in *M. halliana* at day three. Meanwhile, root activity first increased and then decreased, but root pH value showed the opposite trend (Figure 1a,d). FCR increased gradually under Fe deficiency stress (Figure 1e). Jin et al. [33] found a correlation between the number of lateral roots and FCR activity, indicating that an increase in lateral roots contributes to the enhancement of FCR activity induced by Fe deficiency. Under Fe deficiency, plants secrete protons to rhizosphere through plasma membrane P-ATPase to promote Fe dissolution, which is increased by high Fe reductase on epidermal cell membrane, so as to promote Fe bioavailability.

### 4.2. Energy Homeostasis of M. halliana under Fe Deficiency Stress through Carbohydrate Dependent Metabolism

Sugar plays an important role in plant metabolism. It can not only promote the transport and storage of C under abiotic stress [34] but also acts as a compatible protective solute. Sucrose, as a signal molecule regulating biological and abiotic stress response [35], has been observed in Arabidopsis. Here, sucrose accumulation can promote auxin signaling to regulate Fe deficiency and contribute to the tolerance of plants to Fe deficiency caused by leaf chlorosis [36]. In Figure 6, sucrose was significantly up-regulated at the first time point but not changed at the other two time points. The reason is that sucrose can provide energy for plants by degrading and generating glucose and other monosaccharides, so it did not accumulate at the latter two time points. In addition, under the condition of Fe deficiency, higher levels of Glc6P are often used to explain the preparatory steps for increased energy and metabolite demand in the tricarboxylic acid cycle process [37]. In our experiment, the metabolism of digalacturonate, L-xylitol, ribitol, D-xylulose, glucose and glycerol (Figure 6) changed significantly at three time points through the metabolism of glycerides, starch, sucrose, galactose, the carbon fixation of photosynthetic organisms, and glycolysis/gluconeogenesis (Appendix A). Glucose acts as a direct and central signaling molecule in plants. It has been found that glucose can regulate many important processes in photosynthetic plants, such as embryogenesis, seedling development, vegetative growth, reproduction, and senescence [5]. According to a report by Barpeled [38], glucose serves as a carbon source and participates in multiple metabolic pathways (primary and secondary metabolism) in plants. The results showed that, under iron deficiency stress, sucrose content significantly increased. The reason may be the imbalance between the energy demand and supply required for enzyme activity in plants due to iron deficiency [39]. Therefore, the accumulation of sucrose and glucose reflects a defensive strategy for plants to protect themselves in unfriendly environments.

### 4.3. Amino Acid Metabolism, TCA Cycle, and the Response of Secondary Metabolites to Fe Deficiency Stress

Amino acid metabolism has been reported to be closely related to the EMP, PPP, and TCA cycles [40]. In Figure 6, cytosine is metabolized to glutamate and cysteine via glutathione. In recent years, Walch Liu et al. [41] found that glutamate may be used as exogenous signal molecules to regulate the growth and development of roots, leading to more lateral roots. It can synthesize the amino acids needed by plants through transamination. Actually, several studies have shown that the levels of amino acids in plants exhibit different trends in response to different abiotic stresses [42]. Kisaka [43] found that genetically modified tomato fruits contain two to three times more free amino acids and two times more Glu, which is considered a signaling molecule in environmental reactions [44]. L-aspartate and threonine form part of valine through pyruvate, and part of them enter the TCA (Figure 6). Obviously, succinate and fumarate significantly accumulated in R3d-R0d and R3d-R12h, which was due to the insufficient Fe absorption by plant roots under Fe deficiency stress, resulting in the decrease in photosynthetic products in leaves and the acceleration of the energy production of the TCA cycle to maintain the normal growth of *M. halliana* seedlings. Through the comprehensive analysis of primary and secondary metabolites of apple, we can effectively understand the metabolic network of apple under Fe deficiency stress. Meanwhile, the amino acids are used as precursors for numerous metabolites, such as hormones, cell wall components, and a large group of multiple-functional secondary metabolites [45]. Secondary metabolites in plants, such as phenylpropanoids, flavonoids, lignin, monosaccharides, phenolic acids, stilbene, and coumarins, play important roles in biotic and abiotic stress responses and their interactions with the environment [46]. In our study, benzoate accumulated significantly in R3d-R12h, and the longer the stress days were, the more obvious the change in secondary metabolites was. High cellular levels of polyamines correlate with plant tolerance to a wide array of environmental stresses. Moreover, as compared with susceptible plants, stress-tolerant ones generally, in response to abiotic stress, have a large capacity to enhance polyamine biosynthesis [47]. There was a study that demonstrated that Fe deficiency induces putrescine accumulation in roots and that putrescine is involved in the remobilization of Fe from root cell wall hemicellulose through the accumulation of NO. In other words, plants with less endogenous putrescine have less Fe reutilization from the root cell wall and, thus, are more sensitive to Fe deficiency [48]. When plants are iron deficient, they produce a series of metabolites that resist their own growth and maximize their own benefits. In our study, for the first time, changes in various amino acids, phenylpropanoids, and flavonoids were proposed in plant iron deficiency. Especially phenylpropanoids and flavonoids, they are often used in abiotic stress, but plants’ first response to external changes is to produce substances that protect themselves, which will lay the foundation for subsequent research on iron-deficient plants. In addition, our findings can be used for research on iron-deficiency-resistant rootstocks.

## 5. Conclusions

*M. halliana* Fe-deficiency-stress root morphology, pH, and FCR activity were positively correlated with Fe content. Proton secretion from roots increased FCR activity and Fe absorption. Sucrose, as a source of energy, is degraded by glycolysis to produce monosaccharides such as glucose, which accumulate significantly at the first and third time points. Amino acids produce secondary metabolites through the biosynthesis of alkaloids derived from ornithine, lysine, and nicotinic acid and arginine and proline metabolism. Glutamate participates in the TCA cycle through transamination and supplements the photosynthetic products reduced by Fe deficiency stress in *M. halliana*. The increase in putrescine and urea can improve the tolerance of plants to Fe deficiency. In future research, we will focus on studying the metabolic network between carbohydrate metabolism, amino acids, and TCA in *M. halliana* under iron deficiency conditions and screen key genes for cloning and functional analysis.

## Figures and Tables

**Figure 1 plants-13-02500-f001:**
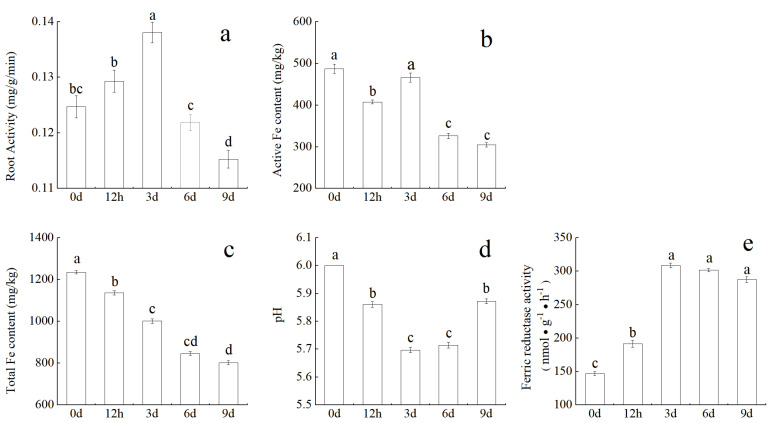
Root parameters of *M. halliana* under Fe deficient stress: (**a**) root active; (**b**) active Fe content; (**c**) total Fe content; (**d**) pH; (**e**) ferric reductase activity. Results are shown as mean (n = 3) ± SE. Different letters indicate statistical differences and different treatments (one-way ANOVA, *p* < 0.05).

**Figure 2 plants-13-02500-f002:**
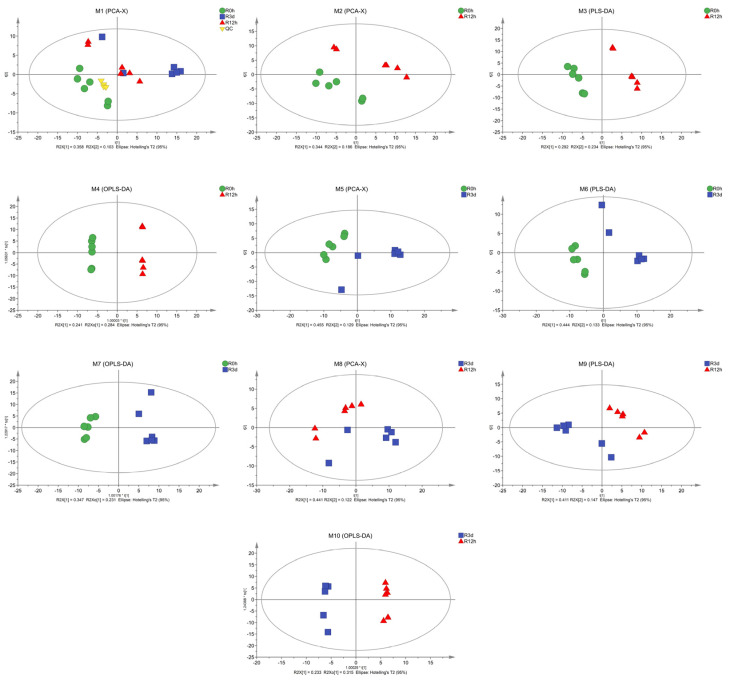
Multivariate statistical analysis showing the metabolomic trajectory of *M. halliana* under Fe-deficient stress. M1 is the principal component analysis (PCA) of all samples; M2, M5, and M8 are PCA in three time points; M3, M6, and least squares d M9 are partial discriminant analysis (PLS-DA); M4, M7, and M10 are orthogonal partial least squares discriminant analysis (OPLS-DA). The horizontal axis of the PCA score plot represents the first principal component, denoted as PC1, using t[1]; the vertical axis represents the second principal component, denoted as PC2, using t[2]. R2X[1] represents the cumulative explained variance of the model in the X-axis direction, while R2X[2] represents the cumulative explained variance of the model in the Y-axis direction.

**Figure 3 plants-13-02500-f003:**
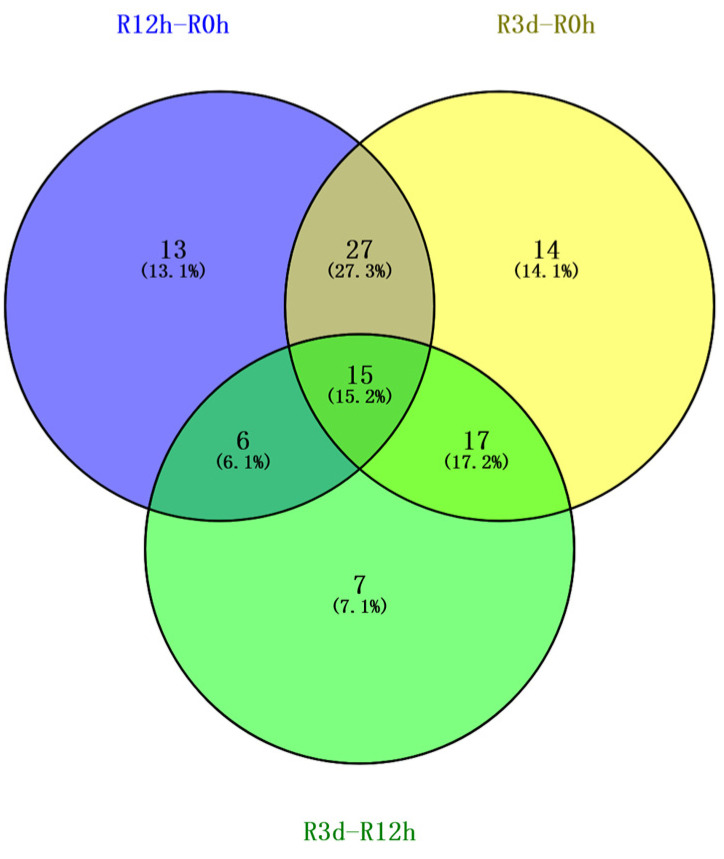
Venn diagram of differential metabolites at three time points under Fe-deficient stress of *M. halliana*. Purple: R12-R0h, yellow: R3d-R0h, green: R3d-R12h. The area of overlap indicates differential metabolites shared by the treatments. The number in each fraction indicates the number of differential metabolites.

**Figure 4 plants-13-02500-f004:**
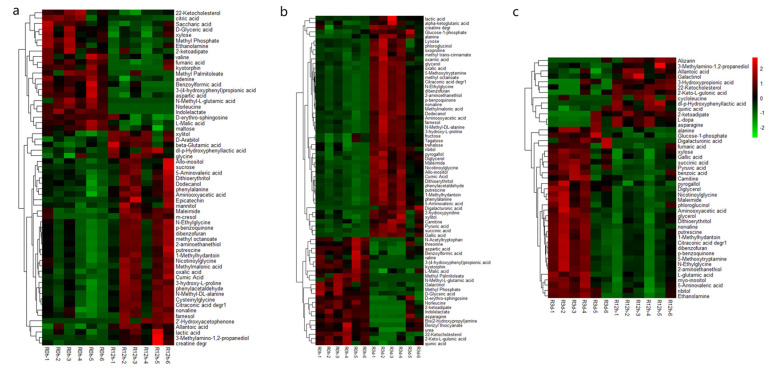
(**a**–**c**) are hierarchical cluster heat maps of differential metabolites at three time points under Fe-deficient stress of *M. halliana* seeding roots. ((**a**): R12h vs. R0h; (**b**): R3d vs. R0h; (**c**): R3d vs. R12h.)

**Figure 5 plants-13-02500-f005:**
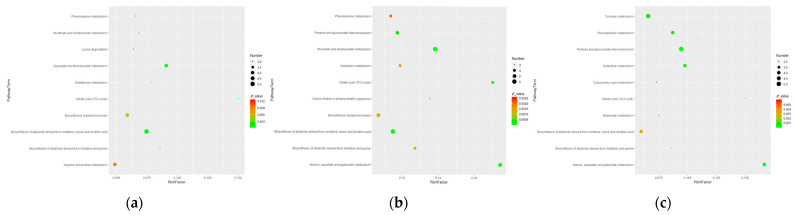
(**a**–**c**) are KEGG pathways of the differential metabolites involved in the response to *M. halliana* at three time points. ((**a**): R12h vs. R0h; (**b**): R3d vs. R0h; (**c**): R3d vs. R12h.)

**Figure 6 plants-13-02500-f006:**
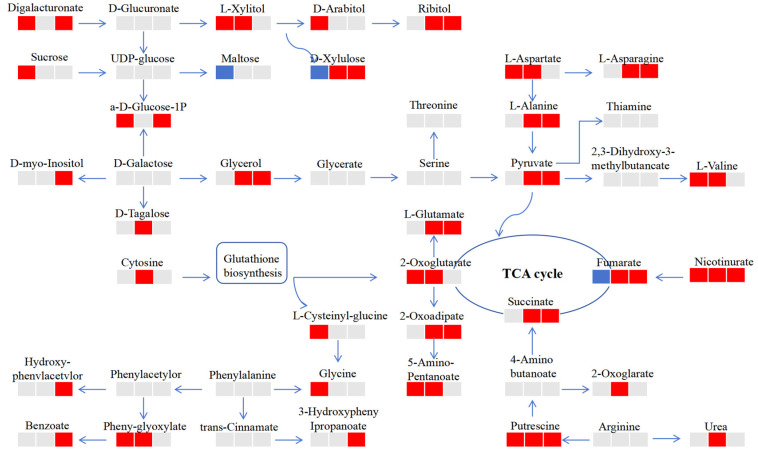
Change in metabolites of the metabolic pathways of *M. halliana* after three time points of Fe-deficient treatment. The metabolites with red boxes denote up-regulation; the metabolites with blue boxes denote down-regulation (*p* < 0.05). Analysis periods include 12 h/0 h; 3 d/0 h; 3 d/12 h. (Metabolic pathway diagram drawn using Adobe Photoshop 2020.)

**Table 1 plants-13-02500-t001:** Effects of Fe-deficient treatment on root morphogenesis of *M. halliana* seedlings. Results are shown as mean (n = 3) ± SE (one-way ANOVA, *p* < 0.05). Different lowercase letters indicate significant differences between treatments.

Stress Time	Total Root Length/cm	Root Surface Area/cm^2^	Average Root Diameter/cm	Total Root Volume/cm^3^	Number of Root Tip	Number of Root Branch
0 d	290.989 ± 8.247 ^e^	14.642 ± 0.294 ^d^	0.430 ± 0.011 ^a^	1.303 ± 0.072 ^d^	1437.667 ± 43.190 ^e^	5819.333 ± 142.956 ^e^
12 h	308.159 ± 8.828 ^d^	15.819 ± 0.034 ^c^	0.436 ± 0.011 ^a^	1.310 ± 0.023 ^d^	1689.333 ± 19.425 ^d^	6618.000 ± 210.231 ^d^
3 d	343.605 ± 6.701 ^c^	16.296 ± 0.285 ^c^	0.439 ± 0.008 ^a^	1.764 ± 0.067 ^c^	2181.333 ± 86.153 ^c^	7302.333 ± 63.311 ^c^
6 d	360.632 ± 4.774 ^b^	17.371 ± 0.095 ^b^	0.435 ± 0.006 ^a^	1.900 ± 0.044 ^b^	2659.000 ± 70.150 ^b^	7768.667 ± 153.468 ^b^
9 d	383.538 ± 5.556 ^a^	18.630 ± 0.517 ^a^	0.449 ± 0.017 ^a^	2.210 ± 0.036 ^a^	3380.667 ± 25.423 ^a^	8453.333 ± 326.944 ^a^

## Data Availability

All data generated or analyzed during this study are included in this published article and its additional files. The collection of samples requires no permissions.

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
