# Peer review of "Metabolism of Malus halliana Roots Provides Insights into Iron Deficiency Tolerance Mechanisms"

_plants, 2024, doi:10.3390/plants13172500_

Round 1

Reviewer 1 Report

Comments and Suggestions for Authors

The presented study contributes to a deeper understanding of plant iron deficiency tolerance mechanisms, offering insights that could be leveraged to improve crop resilience and productivity in iron-deficient soils. The findings reveal that M. halliana roots exhibit increased activity and pH changes under Fe deficiency, alongside alterations in total and effective Fe content and iron reductase (FCR) activity. Metabolomic analysis identified significant changes in metabolites, including sucrose, glucose, and those involved in glutathione metabolism and the TCA cycle, suggesting a coordinated response to Fe deficiency stress. The manuscript will be accepted for publication in Plants after consideration of following points.

Major comments

1. The study mainly investigates the physiological and metabolic responses in apple roots under iron deficiency, which may not fully capture the complexity of plant responses to iron deficiency at the molecular level. Detailed exploration of the specific molecular mechanisms and signaling pathways involved in the plant's response to iron deficiency, could provide a more comprehensive understanding of the physiological changes occurring in plants under stress conditions. Molecular examination is required to validate the current data, gene expression could be studied related to the synthesis or regulation of the major metabolites in response to Fe stress. 

2. The manuscript focuses solely on the response of Malus halliana roots to Fe deficiency stress, without exploring the impact on other parts of the plant. To observe the physiological and developmental changes in plants with altered metabolite levels some growth parameters can be measured to assess plant height, biomass, and root/shoot ratio under stress conditions. Besides, analysis of the concentration of essential nutrients (e.g., nitrogen, phosphorus, potassium) in root tissues is recommended.

3. The size of Table 2 and 3 is too large for the readers to comprehend. In the case of Table 2, the highly expressed (upregulated) metabolites and lower expression (down-regulated) metabolites can be shown in the main text (Table 2), the rest can be added as a supplementary table. Likewise, in the case of Table 3, the most important metabolic pathways based on -log p value in three time points could be presented in the main text (Table 3), the remaining can be shown as a supplementary table. 

Minor comments

Section of Abstract:

1. R0h, R12h, and R3d are mentioned in L18 and 19 without any introduction or explanation of these terms.

2. In L20, first and third time points are mentioned without explaining the chronological time points.

Section of Introduction: 

1. The introduction provides a good overview of iron deficiency in plants. However, it would benefit from a more focused discussion on the unique aspects of Malus halliana and its relevance to the study.

2. Include more recent studies to provide a current perspective on the topic.

3. Mention the research gaps between previous reports and this study. What specific gaps in knowledge does this study aim to address?   

Materials and Methods:

1. In the material and methods section, line no. 102, LC/MS analysis is the sub-heading. Is it correct?  GC/MS was mentioned in the abstract of the manuscript.

Section of Results:

1. There are mistakes in the legend of Figure 1 (1a, b, and c). Please make corrections according to the figure. Moreover, a description of the replication number used, standard error bar, and letters (a, b, c, d) marking significant differences is recommended.

2. Line 166, says there was no significant difference between 0d, 12h and 3d, so the reason for selecting this time point in line 171 is not clear.

3. The legend of Figure 2 should be corrected as M3, M6, and M9 are partial least squares discriminant analysis (PLS-DA) in line no. 194 and 195.

4. The legends in Figure 4 and Figure 5 don’t describe a, b, and c separately. Also, Figure 2, Figure 4 and Figure 5, not clear and fonts too small to read.

5. Table 2, it's better to categorize the metabolites.        

Section of Discussion:

1. Suggest ways in which the insights gained from this study could be used to develop iron-efficient rootstocks.     

Section of Reference:

1. Citation style in text is not correct throughout the whole paper.

2. Citation in line 279 (Jianming et al., 1999) and line 349 (Wittstock and Halkier., 2002) are absent in the reference list.

3. Wei et al. 2008 (line no. 461) should be after Wang et al. 2018 (line no.464) in alphabetical order.

Reviewer 2 Report

Comments and Suggestions for Authors

In the paper entitled “Metabolism of Malus halliana roots provide insights into iron 2 deficiency tolerance mechanisms” authors use a GC-MS-based metabolic approach to identify metabolic pathways affected by iron deficiency in a plant model. The plant used, according to the authors, develops adequately in arid arid soils deficient in this transition metal.

The work contributes to increasing data related to the biochemical mechanisms that plants use to adapt to different environmental conditions; specifically, those pathways that are possibly part of this adaptive response to iron deficiency.

The experiments carried out under controlled conditions allowed us to measure an early response and identify some affected metabolic pathways through statistical analysis adjusted to the data collected.

As for aspects to improve in the written document, the LC/MS subtitle should be corrected by CG-MS and the images should increase the font sizes and resolution.

Author Response

Dear Editors and Reviewers,
Thank you for your kind work and for the reviewers’ comments concerning our manuscript entitled “Metabolism of Malus halliana roots provide insights into iron deficiency tolerance mechanisms” . We have studied comments carefully and have made correction. Revised portion are marked in blue in the paper. The main corrections in the paper and the responses to reviewers’ comments are as flowing:
Reviewer2 :
As for aspects to improve in the written document, the LC/MS subtitle should be corrected by CG-MS and the images should increase the font sizes and resolution.
Response: Thank you for the reviewer's comments. We have corrected the LC/MS subtitles to GC-MS (L129).

Reviewer 3 Report

Comments and Suggestions for Authors

Major revision ,see my comments

Comments on the Quality of English Language

minor editing required 

Round 2

Reviewer 1 Report

Comments and Suggestions for Authors

According as Reviewer's comments, the manuscript has been faithfully corrected to be well improved and now is almost suitable for publication in Plants. One thing that should be pointed out is : Abstract only contains results. It would be better to include a main conclusion at the end of Abstract for readers.

Reviewer 3 Report

Comments and Suggestions for Authors

accepted with minor comments
